

# LINC00958 and HOXC13-AS as key candidate biomarkers in head and neck squamous cell carcinoma by integrated bioinformatics analysis

Dan Xiong[1], Wei Wu[1], Lijuan Kan[1], Dayang Chen[1], Xiaowen Dou[1], Xiang Ji[1], Mengmeng Wang[1], Zengyan Zong[1], Jian Li[2,3] and Xiuming Zhang[1]

[1] Medical Laboratory of the Third Affiliated hospital of ShenZhen University, Shenzhen, China
[2] Department of Otolaryngology, The First Affiliated Hospital, Sun Yat-sen University, Guangzhou, China
[3] Guangzhou Key Laboratory of Otorhinolaryngology, Guangzhou, China

Corresponding authors
Jian Li, lijianent@hotmail.com
Xiuming Zhang, zxm0760@163.com

## ABSTRACT

**Background.** Head and neck squamous cell carcinoma (HNSCC) is a malignant tumor with a strong tendency for metastasis and recurrence. Finding effective biomarkers for the early diagnosis of HNSCC is critical for the early treatment and prognosis of patients.

**Methods.** RNA sequencing data including long non-coding RNAs (lncRNAs), messenger RNA (mRNAs) and microRNAs (miRNAs) of 141 HNSCC and 44 adjacent normal tissues were obtained from the TCGA. Differentially expressed genes were analyzed using the R package DESeq. GO terms and Kyoto Encyclopedia of Genes and Genomes (KEGG) pathway enrichment analyses were conducted. A competing endogenous RNAs (ceRNA) network was constructed. The most differentially expressed genes in the main ceRNA network were chosen for nasopharyngeal carcinoma (NPC) cell lines and NPEC2 Bmi-1 cell line verification. A receiver operating characteristic (ROC) curve was constructed for 141 specimens of HNSCC tissues from 44 control samples.

**Results.** In our study, 79 HNSCC-associated abnormally expressed lncRNAs, 86 abnormally expressed miRNAs and 324 abnormally expressed mRNAs were identified. The public microarray results showed that LINC00958 and HOXC13-AS expression levels were upregulated in HNSCC tissues compared with the adjacent normal tissues in this study ($p < 0.0001$). LINC00958 and HOXC13-AS expression levels in NPC cell lines were higher than those in the NPEC2 Bmi-1 cell line ($p < 0.05$). The results showed that the area under the ROC curve (AUC) of LINC00958 reached up to 0.906 at a cutoff value of 7.96, with a sensitivity and specificity of 80.85% and 90.91%, respectively. The AUC of HOXC13-AS reached up to 0.898 at a cutoff value of 0.695, with sensitivity and specificity values of 86.23% and 83.78%, respectively.

**Conclusion.** The current study indicates that LINC00958 and HOXC13-AS are new candidate diagnostic biomarkers for HNSCC patients.

## INTRODUCTION

Head and neck squamous cell carcinoma (HNSCC) includes a group of tumors arising in the oral cavity, oropharynx, nasal cavity and paranasal sinuses, larynx and hypopharynx (*Economopoulou et al., 2019*). HNSCC was reported as the sixth most common type of malignancy in human (*Wu et al., 2019*). Estimately 50,0000 new cases are being diagnosed annually (*Ghafouri-Fard et al., 2019*). Despite the progress made in overall therapy in recent years, the five-year overall survival rate for HNSCC patients is only about 50% (*Cossu et al., 2019*; *Ren et al., 2016*). The poor prognosis of HNSCC patients are mainly because of local invasions, treatment-resistance, recurrence, and metastasis (*Ren et al., 2016*; *Yang et al., 2019*). Patients with recurrence or metastasis after treatment often have a poor prognosis, which is the main reason for the failure of treatment and the poor survival rate of HNSCC patients. Therefore, screening biomarkers of HNSCC for early detection, the prediction of prognosis and the monitoring of recurrence is of great significance for the clinical diagnosis and treatment of HNSCC.

To improve the diagnosis and prognosis of HNSCC, novel effective and sensitive biomarkers are needed. lncRNAs are a kind of noncoding RNA with a transcript length of more than 200 bp. Their important regulatory role in tumorigenesis and metastasis has been a research focus in recent years. Although the clinical significance of lncRNAs and their function and mechanism in cancer are still unclear, they are involved in cancer progression and development beyond all doubt. Studies have indicated that lncRNAs can serve as potential biomarkers for the early diagnosis and prognosis of multiple cancers, including HNSCC (*Guglas et al., 2017*; *Li et al., 2019a*). Recent reports have indicated that lncRNAs have multiple biological functions in HNSCC, including signaling regulation, invasion, metastasis, and potential prognostic biomarkers (*Xiong et al., 2019*; *Zhang et al., 2019b*). *Qiu et al. (2019)* found that knockdown of RHPN1-AS1 inhibited cell migration, invasion and proliferation of HNSCC, indicating that RHPN1-AS1, acting as an oncogene, may be a potential diagnostic and therapeutic target in HNSCC. *Jiang et al. (2019)* suggested that LINC00460 facilitated PRDX1 entry into the nucleus and promoted EMT in HNSCC cells. LINC00460 and PRDX1 can serve as both promising candidate prognostic predictors and potential targets for cancer therapy for HNSCC. The prognostic and diagnostic value of lncRNAs in HNSCC should not be underestimated, and their clinical significance will allow patients with HNSCC to receive more personalized treatment.

In the present study, The Cancer Genome Atlas (TCGA) database was used to identify differentially expressed genes (lncRNAs, miRNA and mRNA) between HNSCC and adjacent normal tissues. Gene Ontology (GO) terms and Kyoto Encyclopedia of Genes and Genomes (KEGG) pathway enrichment analyses were conducted to further understand the molecular functions of mRNAs. A ceRNA coexpression network of HNSCC was constructed to help clarify the functions of the noncoding RNAs in HNSCC. Nasopharyngeal carcinoma (NPC) is a type of HNSCC with a remarkable regional and ethnic specificity that can lead to serious health problems in south China and southeastern Asia compared with the Western world (*Economopoulou et al., 2019*). To screen candidate biomarkers from these differentially expressed genes, particularly those in the ceRNA network, we performed

**Table 1** The pathological stage information of head and neck squamous cell carcinoma tissues and adjacent normal tissues from TCGA database in this study.

| Adjacent normal tissues from HNSCC patients | | HNSCC tumor tissues | |
|---|---|---|---|
| **Stage** | **Number** | **Stage** | **Number** |
| NA | 1 | NA | 10 |
| stage I | 2 | stage I | 15 |
| stage II | 16 | stage II | 40 |
| stage III | 8 | stage III | 30 |
| stage IVa | 17 | stage IVa | 43 |
| | | stage IVb | 2 |
| | | stage IVc | 1 |

**Table 2** The origins information of head and neck squamous cell carcinoma tissues and adjacent normal tissues from TCGA database in this study.

| | HNSCC | Normal |
|---|---|---|
| Lip | 2 | 0 |
| Oropharynx | 2 | 0 |
| Base_of_Tongue | 3 | 2 |
| Hypopharynx | 3 | 0 |
| Hard_Palate | 4 | 0 |
| Alveolar_Ridge | 5 | 0 |
| Tonsil | 6 | 0 |
| Buccal_Mucosa | 10 | 0 |
| Floor_of_Mouth | 19 | 3 |
| Oral_Cavity | 21 | 14 |
| Larynx | 23 | 12 |
| Oral_Tongue | 43 | 13 |

real-time PCR on NPC and NPEC2 Bmi-1 cells lines. Finally, we identified two lncRNAs, LINC00958 and HOXC13-AS, as new candidate biomarkers for HNSCC patients.

## MATERIAL AND METHODS

### HNSCC patients and RNA expression profiles

The RNA sequencing data of HNSCC and normal control samples were acquired in The Cancer Genome Atlas (TCGA, https://cancergenome.nih.gov/) database. Firstly, we excluded patients that were firstly histologic diagnosed with non-HNSCC, Furthermore, patients with incomplete data information such as clinicopathologic data, or the presence of another malignancy besides HNSCC, or received chemotherapy or radiotherapy were excluded. Finally, a total of 141 HNSCC and 44 normal control samples contained mRNA, miRNA and lncRNA data were analyzed in this study. The complete sample information is in Supplement Table S1. Tables 1–3 show the tumor stage, origins, and institutions

**Table 3** The institutions information of head and neck squamous cell carcinoma tissues and adjacent normal tissues from TCGA database in this study.

| Institutions | Number |
| --- | --- |
| Duke University | 1 |
| Fox Chase | 1 |
| ILSBio | 1 |
| Medical College of Georgia | 1 |
| Memorial Sloan Kettering Cancer Center | 1 |
| Molecular Response | 1 |
| St. University of Colorado Denver | 1 |
| University Of Michigan | 1 |
| ABS - IUPUI | 2 |
| Barretos Cancer Hospital | 2 |
| Fred Hutchinson | 2 |
| Johns Hopkins | 2 |
| Montefiore Medical Center | 2 |
| University of Minnesota | 2 |
| University of Schleswig-Holstein | 2 |
| International Genomics Consortium | 3 |
| University of Miami | 3 |
| Emory University—Winship Cancer Inst. | 4 |
| Asterand | 5 |
| UNC | 6 |
| Greater Poland Cancer Center | 10 |
| Vanderbilt University | 13 |
| University Health Network, Toronto | 19 |
| University of Pittsburgh | 21 |
| MD Anderson Cancer Center | 79 |

information of these samples. Figure 1 shows the flow chart for bioinformatics analysis performed in this study.

## Cell culture

NPEC2 Bmi-1 cells grown in keratinocyte/serum-free (KSF) medium (Invitrogen) are immortalized nasopharyngeal epithelial cells induced by Bmi-1 as described previously (*Zhang et al., 2018*). All human nasopharyngeal carcinoma cell lines (SUNE1, SUNE2, 6-10B, 5-8F, CNE2 and HONE1) were maintained in our laboratory and cultured in RPMI 1640 medium (GIBCO) supplemented with 5% fetal bovine serum (FBS, GIBCO). All cells were cultured in a humidified 5% $CO_2$ incubator at 37 °C.

## Quantitative real-time PCR

Total RNA was extracted from cultured cells using Trizol reagent (Invitrogen, Grand Island, NY) and reversely transcribed using a reverse transcriptase kit (Invitrogen) according to the manufacturer's instructions. The cDNA was then used as template in the real-time PCR using Power SYBR Green qPCR SuperMix-UDG (Invitrogen). The procedure of the

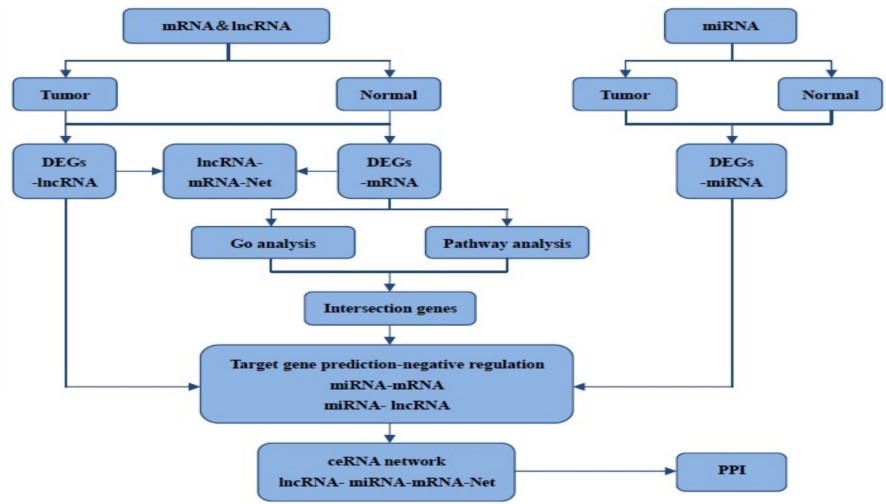

**Figure 1  The flow chart for bioinformatics analysis in this study.**

real-time PCR reaction as follow: (1) 50 °C for 2 min; (2) pre-denaturation at 95 °C for 10 min; (3) denaturation at 95 °C for 15 s, annealing and extension at 60 °C for 1 min, 40 cycles. The real-time PCR reaction was carried out on an ABI PRISM 7500 Sequence Detection System (Applied Biosystems, Foster City, CA). All gene expression levels were normalized to that of the housekeeping gene GAPDH, which was used as an internal standard. The forward primer for LINC00958 was 5′-GCCTGGCACATTCAGTGGAGAG-3′, and the reverse primer was GTGGCGGCCTGAGCTTCTTC. The forward primer for HOXC13-AS was 5′-CCTCAAGAAGACCAGCCGAAGTTG-3′, and the reverse primer was ATTGTTCAGAGCAAGCGGACTTCC. The forward primer for GAPDH was 5′-CGAGGTCATAGTTCCTGTTGGTG-3′, and the reverse primer was CCCAATACGACCAAATCCGTT.

## Data processing and differential expression analysis

Samples were divided into tumor tissues vs. adjacent nontumor tissues to detect the differentially expressed genes (DEGs) of lncRNAs, mRNAs and miRNAs. The raw RNA sequencing profile was postprocessed and normalized by the TCGA RNASeqV2 system. No further normalization was required. Ensemble gene IDs (ENSGs) were mapped to gene symbols and gene types (protein-coding genes or noncoding genes) according to the annotation files in the GENCODE database (version 22). The DEGs between the NPC and normal samples were analyzed using the R package DESeq (http://www.bioconductor.org/packages/release/bioc/html/DESeq.html). A |log fold-change (FC)|≥ 2 and a false discovery rate (FDR) <0.01 were set as the cut-off criteria.

## Functional and pathway enrichment analyses

To further understand the biological processes, molecular functions and signaling pathways of the protein-coding genes (mRNAs) among the differentially expressed RNAs, GO terms and KEGG pathway enrichment analyses were performed with the help of the Database for

Annotation, Visualization and Integrated Discovery (DAVID, https://david.ncifcrf.gov/).
Up- and down-regulated mRNAs were analyzed. The thresholds were set as a $p$ value <0.05 and a false discovery rate (FDR) <0.01.

### ceRNA and PPI networks

Based on the relationship among the lncRNAs, miRNAs and mRNAs, a lncRNA-miRNA-mRNA ceRNA network was built. miRanda (http://www.microrna.org/) and TargetScan (http://www.targetscan.org/) were used to predict the miRNA–mRNA and miRNA–lncRNA interactions. According to the theory that lncRNAs can regulate the activities of mRNAs by acting as miRNA sponges, the mRNAs with negatively regulated lncRNAs and miRNAs were selected to construct the lncRNA-miRNA-mRNA ceRNA network. Only the lncRNAs, miRNAs, and mRNAs with an upregulated fold change >1.5 or a downregulated fold change <0.50 and a $p$ value <0.05 were retained. Afterward, in the context of the coexpressed genes, the PPI network was constructed via STRING (Version 10.5) (https://string-db.org/).

### Statistical analysis

Statistical analyses were performed using the statistical software package SPSS 16.0. Differences among variables were analyzed by 2-tailed Student's t tests. Data are presented as the mean ± SD unless otherwise indicated. $p$ values ≤0.05 were considered statistically significant.

## RESULTS

### Differentially expressed lncRNAs, microRNAs and mRNAs in HNSCC and adjacent normal tissues from the TCGA database

In this study, 141 HNSCC patient tissues and 44 adjacent normal tissues were selected from the TCGA database. Then, the lncRNA, miRNA and mRNA expression of 141 HNSCC tissues and 44 adjacent normal tissues was compared to identify the significantly different RNAs through certain criteria (upregulated fold change >1.5 or downregulated fold change <0.50 and $p < 0.05$). Then, 79 HNSCC-associated abnormally expressed lncRNAs (37 upregulated and 42 downregulated), 86 abnormally expressed miRNAs (36 upregulated and 50 downregulated) and 324 abnormally expressed mRNAs (158 upregulated and 166 downregulated) were identified. Figure 2A shows the top 15 upregulated lncRNAs and the top 15 downregulated lncRNAs. Figure 2B shows the top 15 upregulated miRNAs and the top 15 downregulated miRNAs, and Fig. 2C shows the top 15 upregulated mRNAs and the top 15 downregulated mRNAs.

### Gene Ontology and pathway analyses in HNSCC and adjacent normal tissues from the TCGA database

Finally, both the up- and downregulated mRNAs were further analyzed using the GO database (http://www.geneontology.org). These differentially expressed genes represent a measure of significance of a function. For instance, for the upregulated mRNAs, the most enriched GO term was "signal transduction", and other significant GO terms included "extracellular matrix organization", "cell adhesion", and "neutrophil degranulation". For

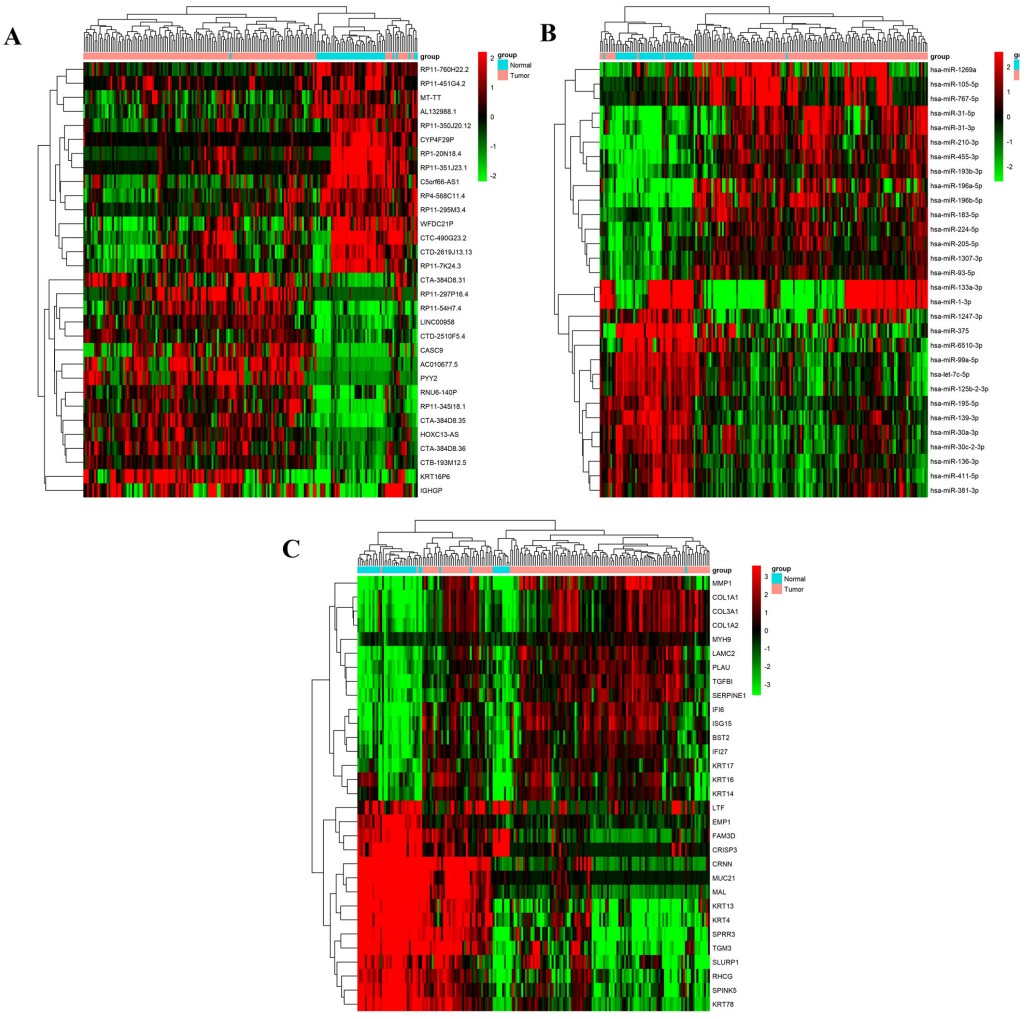

**Figure 2** **Cluster analysis of 15 up- and down-regulated differentially expressed lncRNAs, microRNA and mRNAs in HNSCC tissues and adjacent normal tissues from TCGA database.** Hierarchical clustering analysis indicated that 15 up- and down-regulated differentially expressed lncRNAs (A), microRNA (B) and mRNAs (C) were differentially expressed between HNSCC patients and healthy controls, respectively. The red and the green shades represent the expression levels above and below the relative expression among all samples.

the downregulated mRNAs, the most enriched GO term was "neutrophil degranulation" (Figs. 3A–3B). These data provided a definitive functional description of the genes differentially expressed in HNSCC. We next subjected these differentially expressed genes to pathway analysis, where the most enriched network corresponding to the upregulated transcripts was "human papillomavirus infection", and the most enriched network corresponding to the downregulated transcripts was "metabolic pathways" (Figs. 3C–3D).

## ceRNA and PPI network construction

To establish the lncRNA-miRNA-mRNA ceRNA network, lncRNAs and mRNAs targeted by miRNAs were identified from the above data. A total of 38 miRNAs, 66 lncRNAs and

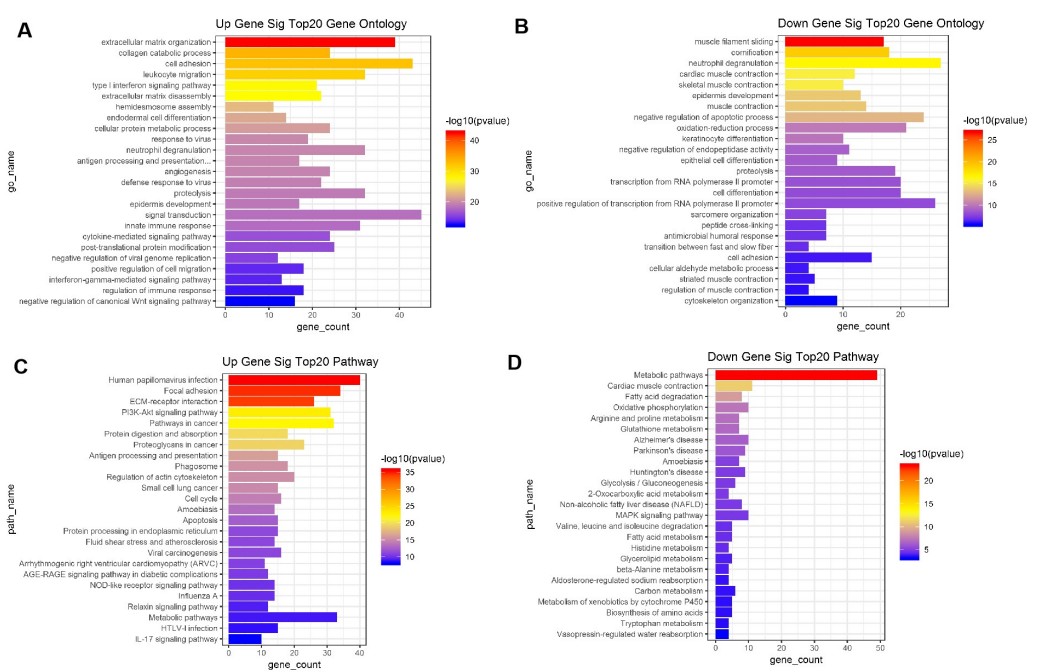

**Figure 3** **The 20 key gene ontology (GO) terms and KEGG pathway enrichment of differentially expressed intersection mRNAs for HNSCC groups versus adjacent normal controls.** (A) The 20 key Gene Ontology terms in up-regulated differentially expressed genes, the bar plot shows the enrichment scores of the significant enrichment GO terms. (B) The 20 key Gene Ontology terms in down-regulated, differentially expressed genes, the bar plot shows the enrichment scores of the significant enrichment GO terms. (C) The 20 key pathways in up-regulated differentially expressed genes, the bar plot shows the enrichment scores of the significant enrichment pathways. (D) The 20 key pathways in down-regulated differentially expressed genes. The bar plot shows the enrichment scores of the significant enrichment pathways.

102 mRNAs were included in the ceRNA network (Fig. 4). Seventy-two upregulated and 19 downregulated mRNAs were included in the PPI network (Fig. 5). Proteins with high degrees were classified as hub proteins. The most significant hub proteins were HSP90AA1, CD44, TOP2A and ITGAV.

## LINC00958 and HOXC13-AS expression levels in tumors and cells

The long noncoding RNA LINC00958 has been reported to facilitate cell proliferation and migration in oral squamous cell carcinoma (*Wang, Zheng & Ren, 2019*). HOXC13-AS was confirmed to positively affect cell proliferation and invasion in nasopharyngeal carcinoma (*Dai et al., 2019*). Our results showed that LINC00958 and HOXC13-AS expression levels were upregulated in HNSCC tissues compared with that in the adjacent normal tissues in this study ($p < 0.0001$, Figs. 6A–6B). To validate the expression levels of LINC00958 and HOXC13-AS identified through our bioinformatics analyses, real-time PCR was performed to detect the expression levels of LINC00958 and HOXC13-AS in NPC cell lines and in the NPEC2 Bmi-1 cell line. LINC00958 and HOXC13-AS expression levels in the NPEC2 Bmi-1 cell line were lower than those in the NPC cell lines ($p < 0.0001$, Figs. 6C–6D). ROC curve was constructed for 141 specimens of HNSCC tissues from 44 control samples. The

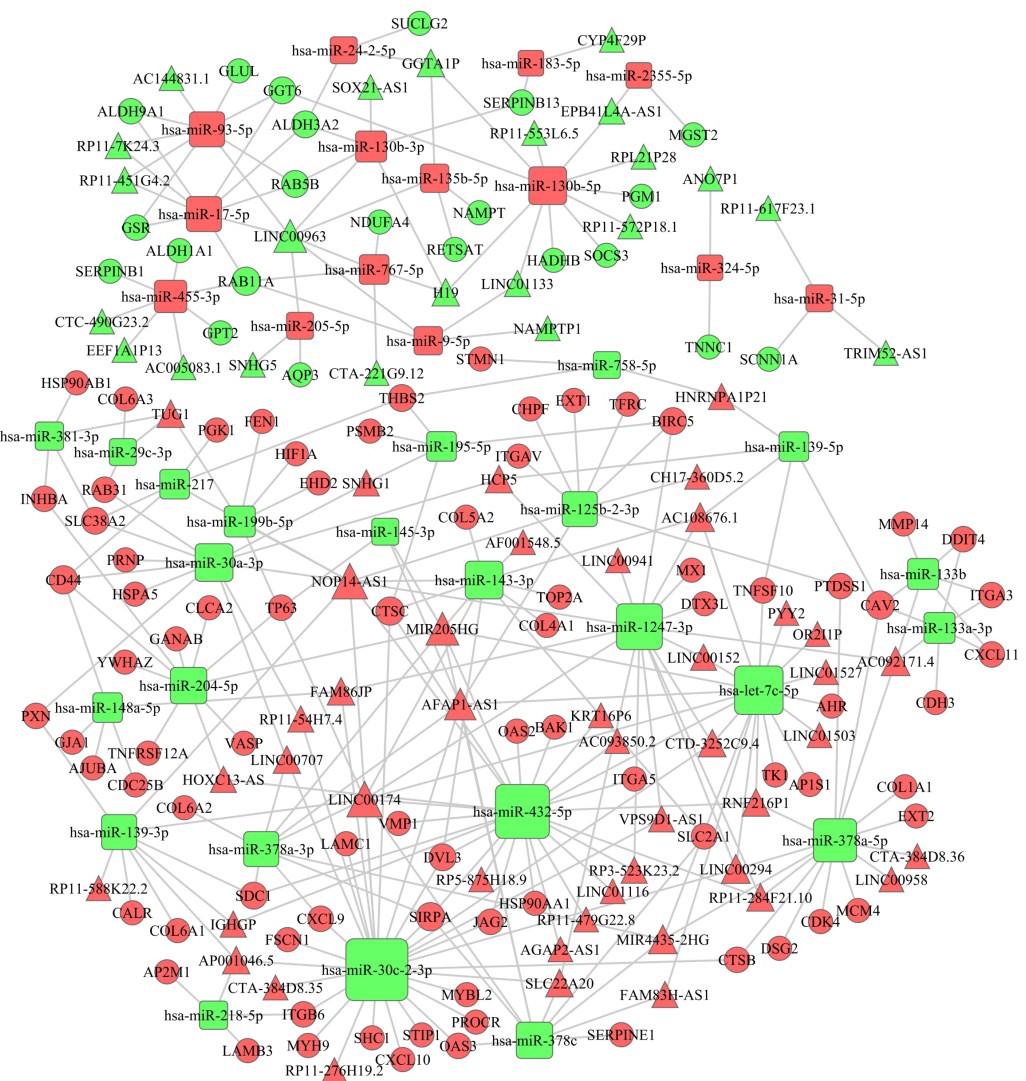

**Figure 4 The ceRNA network of lncRNA–miRNA–mRNA in HNSCC.** The red represents the upregulated, and the green represents the downregulated. Diamonds represent miRNAs, balls represent mRNAs, and triangles represent lncRNAs.

results showed that the area under the ROC curve (AUC) of LINC00958 reached up to 0.906 (95% CI [0.863–0.949]; $p < 0.001$, Fig. 6E). The optimal cutoff value was 7.96, with a sensitivity and specificity of 80.85% and 90.91%, respectively. The results showed that the AUC of HOXC13-AS reached up to 0.898 (95% CI [0.834–0.962]; $p < 0.001$, Fig. 6F). The optimal cutoff value of HOXC13-AS expression was 0.695, with sensitivity and specificity values of 86.23% and 83.78%, respectively.

## DISCUSSION

The TCGA is a comprehensive bioinformatics database that contains information on various malignant tumors. Based on the TCGA bioinformatics analysis, many studies have
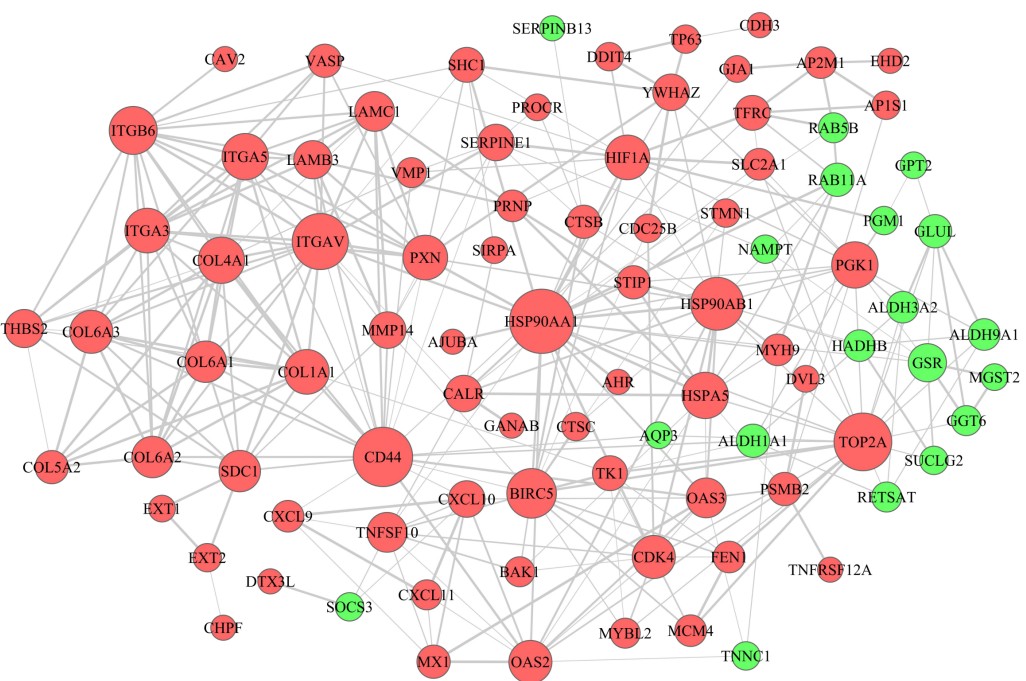

**Figure 5** **PPI network of 72 upregulated and 19 downregulated DEGs.** Nodes represent proteins and edges represent interactions between two proteins. Red nodes represent upregulated DEGs and green nodes represent downregulated DEGs.

been carried out to improve our understanding of the pathological process of multiple cancers. Through an integrative analysis of GEO and TCGA data, *Zhang et al. (2019a)* suggested that three lncRNAs, UCA1, HOTTIP, and HMGA1P4, may contribute to the development of gastric cancer and may be related to the prognosis of gastric cancer patients. *Wang, Zheng & Ren (2019)* used bioinformatics and showed that TRMT12 might involve in the progression and metastasis of HNSCC, and could be served as an independent biomarker of poor prognosis in HNSCC. *Dai et al. (2019)* found that HOXC10 facilitated WNT-dependent epithelial-mesenchymal transition (EMT) and might be a potential prognostic biomarker and a therapeutic target in oral squamous cell carcinoma (OSCC). In our study, the AUC of LINC00958 reached up to 0.906. The optimal cutoff value was 7.96, with a sensitivity and specificity of 80.85% and 90.91%, respectively. The results showed that the AUC of HOXC13-AS reached up to 0.897. The optimal cutoff value of HOXC13-AS expression was 0.695, with sensitivity and specificity values of 86.23% and 83.78%, respectively. Finally, we identified two lncRNAs, LINC00958 and HOXC13-AS, as new candidate biomarkers for HNSCC patients by bioinformatics analysis.

Several studies have suggested that LINC00958 and HOXC13-AS are significantly associated with tumor progression, migration, and invasion. A recent report showed that LINC00958 regulated the miR-627-5p/YBX2 axis and facilitated cell proliferation and migration in oral squamous cell carcinoma (*Chen et al., 2019a*). Wen et al. reported that

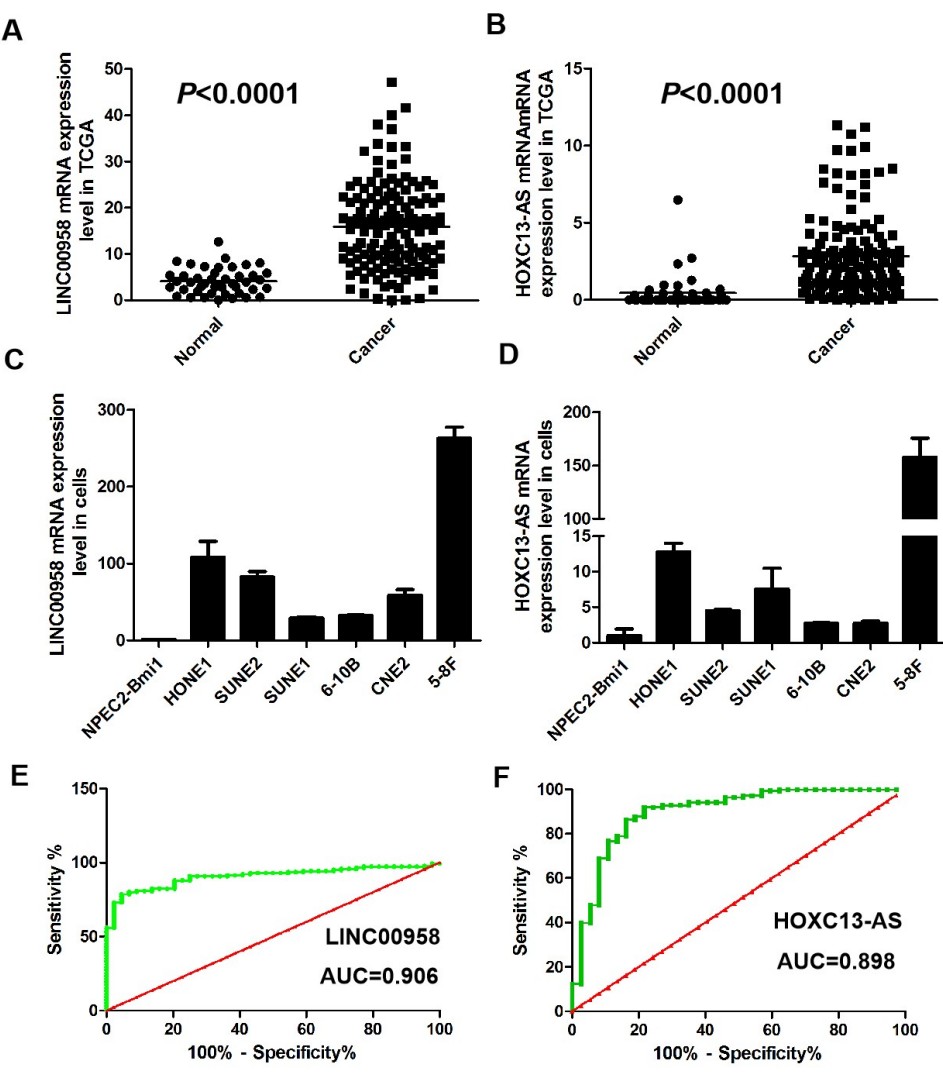

**Figure 6** **LINC00958 and HOXC13-AS expression levels in tumors and cells.** (A–B) Enhanced expression of LINC00958 and HOXC13-AS expression levels in HNSCC tissues compared with adjacent normal tissues ($n = 44$, $T = 141$, $p < 0.0001$). (C-D) LINC00958 and HOXC13-AS expression levels in NPC cell lines were higher than that in NPEC2 Bmi-1 which was immortalized nasopharyngeal epithelial cells induced by Bmi-1. (E–F) The ROC curve of LINC00958 and HOXC13-AS between HNSCC tissues and controls.

silencing LINC00958 prevented tumor initiation by acting as a sponge of microRNA-330-5p to downregulate PAX8 in pancreatic cancer (*Chen et al., 2019b*). *Seitz et al. (2017)* showed that LINC00958 was significantly upregulated in bladder cancer and exhibited phenotypic functions that showed carcinogenic characteristics. A recent study reported that HOXC13-AS was obviously upregulated in breast cancers and promoted cell proliferation by regulating the miR-497-5p/PTEN axis (*Li et al., 2019b*). HOXC13-AS positively affects cell proliferation and invasion by modulating the miR-383-3p/HMGA2 axis in NPC

(*Gao et al., 2019*). In our study, there was an obvious increase in LINC00958 and HOXC13-AS expression at the mRNA level between HNSCC and adjacent tissues. Next, the exact function and mechanism of LINC00958 and HOXC13-AS in HNSCC were further explored, including proliferation, invasion and metastasis.

The differentiation and progression of HNSCC cells may be closely related to the regulation of certain genes. In our study, GO and pathway enrichment analyses revealed that the upregulated factors are mainly involved in 'extracellular matrix organization', 'human papillomavirus infection', 'focal adhesion', 'ECM-receptor interaction', 'PI3K-Akt signaling pathway', and 'pathway cancer', while the downregulated factors are associated with 'muscle filament sliding', 'cornification', 'metabolic pathways', 'cardiac muscle contraction' and 'fatty acid degradation'. Extracellular matrix organization and cell adhesion play an important role in the invasion and metastasis of HNSCC cells.

Although this study has demonstrated that LINC00958 and HOXC13-AS are useful in clinical discrimination of HNSCC and adjacent tissues, several challenges remain. In the TCGA base squamous cell carcinoma are arising from different origins. Then the LINC00958 and HOXC13-AS expression value in different origins of HNSCC were compared using the Kruskal–Wallis Test for origin with a frequency of 10 or more. The results were showed in Table S2. At p 0.05 significance level, we concluded that there is no difference between different origins of squamous cell carcinoma. Figures S1 and S2 showed LINC00958 and HOXC13-AS expression value in different origins respectively. Another challenge is to obtain more clinical samples validating results, especially those from nasopharynx site.

In conclusion, our study revealed differentially expressed lncRNAs, microRNAs and mRNAs between nasopharyngeal carcinoma and adjacent normal tissues from the TCGA database. Different Gene Ontology terms, pathways, and ceRNA and PPI networks were obtained. We identified two lncRNAs, LINC00958 and HOXC13-AS, as new candidate biomarkers for NPC patients by bioinformatics analysis. These results indicate that the transcript levels of LINC00958 and HOXC13-AS may be suitable biomarkers of NPC.

### Funding

This study was supported by the National Natural Science Foundation of China (Grant NO. 81772921, 81502344), a grant from the Science and Technology Planning Project of Shenzhen City of China (NO. JCYJ20180306172209668), the discipline construction ability promotion project of Shenzhen health and population family planning commission (NO. SZXJ2017018), the Sanming Project of Medicine in Shenzhen (NO. SZSM201601062), the Natural Science Foundation of Guangdong Province (NO. 2016A030313257), and the Science and Technology Planning Project of Guangdong Province (NO.2014A020212141). The funders had no role in study design, data collection and analysis, decision to publish, or preparation of the manuscript.

## Grant Disclosures

The following grant information was disclosed by the authors:

National Natural Science Foundation of China: 81772921, 81502344.

Science and Technology Planning Project of Shenzhen City of China: JCYJ20180306172209668.

Discipline construction ability promotion project of Shenzhen health and population family planning commission: SZXJ2017018.

Sanming Project of Medicine in Shenzhen: SZSM201601062.

Natural Science Foundation of Guangdong Province: 2016A030313257.

Science and Technology Planning Project of Guangdong Province: 2014A020212141.

## Competing Interests

The authors declare there are no competing interests.

## Author Contributions

- Dan Xiong conceived and designed the experiments, prepared figures and/or tables, and approved the final draft.
- Wei Wu and Zengyan Zong performed the experiments, analyzed the data, prepared figures and/or tables, and approved the final draft.
- Lijuan Kan performed the experiments, analyzed the data, prepared figures and/or tables, authored or reviewed drafts of the paper, and approved the final draft.
- Dayang Chen, Xiaowen Dou, Xiang Ji and Mengmeng Wang performed the experiments, analyzed the data, authored or reviewed drafts of the paper, and approved the final draft.
- Jian Li and Xiuming Zhang conceived and designed the experiments, analyzed the data, prepared figures and/or tables, and approved the final draft.

## Data Availability

The raw measurements are available in the Supplemental Files.

## Supplemental Information

Supplemental information for this article can be found online at http://dx.doi.org/10.7717/peerj.8557#supplemental-information.

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
