# Peer review of "LINC00958 and HOXC13-AS as key candidate biomarkers in head and neck squamous cell carcinoma by integrated bioinformatics analysis"

_PeerJ, doi:10.7717/peerj.8557_

## Round 0.1 · original submission · Major Revisions

Two specialists in the field evaluated the present manuscript, and both have concerns related to this paper. The reviewers have described the significant points that should be answered by the authors. Considering the evaluation carried out by both reviewers, I recommend major revision in this submission.

Reviewer 1 ·

Basic reporting

1) Abstract has some cell line acronyms like NPEC2 25 Bmi-1, would be helpful to names of them. Some couple gene symbols need also to be elaborated.

2) References seems to be old, can you replace them with last 10 years references?

3) English and grammar seems reasonable to me.

Experimental design

1) Authors state that "In the present study, The Cancer Genome Atlas (TCGA) database was used to identify 68 differentially expressed genes (lncRNAs, miRNA and mRNA) between HNSCC and adjacent 69 normal tissues." Why this number sample size chosen?

2) Where the samples from same batch?

3) I don't see sample ids from TCGA database, can authors please provide them?

4) Cluster analysis for heatmaps looks crowded, can authors separate them into different figures instead of different panel?

5) The threshold of >1.5 was chosen significant, can authors explain the reason or the reference to do that?

6) Can authors validate expression of two significant genes with RT PCR if access to tissue samples?

Validity of the findings

The data on which the conclusions are based in an acceptable discipline-specific repository.

Reviewer 2 ·

Basic reporting

Dear authors,
Thank you for your hard work with this paper.
New biomarkers in SCC are strongly needed to be discovered.
The English language is fluent. However you should change some abbreviations.
In literature there are head and neck cancers you should remove headneck phraze and change.
Also you neet to use one type of abbreviation (title, 16,41).
You should check references list and improve (e.g.221-222).
Raw data and figures are good shared.
Differences in title should be made, because you are testing only nasopharyngeal cell lines, so it have to be mentioned.also source of both cell lines is missing.
Please change manuscritp and write about nasopharyngeal cancers according to WHO guidance.

Experimental design

In the TCGA base there are tissues from oral cavity, oropharynx and laryngeal sites ( Nature volume 517, pages 576–582 (29 January 2015), but aren’t any from nasopharyngeal site. You have to mention about the orgin/location of tested squamous cell carcinomas and cell lines.
There is no marker to divide if squamous cell carcinoma is arising from lung or head and neck region, however in is unclear if there are different mechanisms of RNA action, please mention that.
Please write about orgin of cell lines.

Validity of the findings

Findings are interesting, but it is unclear if you can compare nasopharyngeal orgin with other orgins of squamous cell carcinoma.
What would you think, would be appropriate to compare squamous cell carcinoma of lung and head and neck due to lnc/micRNA ?
Even during clinical trials nasopharyngeal cancers are excluded from treatment for other locations.

---

## Round 0.2 · accepted · Accept

The authors performed extensive revision of the manuscript as indicated by the reviewers. In my view, the manuscript improved a great deal and can be accepted as it is.